# Site-selective remote C(sp³)–H heteroarylation of amides via organic photoredox catalysis

Hui Chen[1], Wenjing Fan[2], Xiang-Ai Yuan (ID) [2]* & Shouyun Yu (ID) [1]*

Radical translocation processes triggered by nitrogen-centered radicals (NCRs), such as 1,5-hydrogen atom transfers (1,5-HAT), demonstrated by the well-established Hofmann-Löffler-Freytag (HLF) reaction, provide an attractive approach for the controllable and selective functionalization of remote inert C(sp³)–H bonds. Here we report an amidyl radical-triggered site-selective remote C(sp³)–H heteroarylation of amides under organic photoredox conditions. This approach provides a mild and highly regioselective reaction affording remote C(sp³)–H heteroarylated amides at room temperature under transition-metal free, weakly basic, and redox-neutral conditions. Non-prefunctionalized heteroarenes, such as purines, thiazolopyridines, benzoxazole, benzothiazoles, benzothiophene, benzofuran, thiazoles and quinoxalines, can be alkylated directly. Sequential and orthogonal C–H functionalization of different heteroarenes by taking advantage pH value or polarity of radicals has also been achieved. DFT calculations explain and can predict the site-selectivity and reactivity of this reaction. This strategy expands the scope of the Minisci reaction and serves as its alternative and potential complement.

---

[1] State Key Laboratory of Analytical Chemistry for Life Science, Jiangsu Key Laboratory of Advanced Organic Materials, School of Chemistry and Chemical Engineering, Nanjing University, Nanjing 210023, China. [2] School of Chemistry and Chemical engineering, Qufu Normal University, Qufu 273165, China. *email: xiang_aiyuan@126.com; yushouyun@nju.edu.cn

Heterocycles are important structural motifs in a large number of natural products, and in multiple commercial products, such as advanced materials and pharmaceuticals[1–3]. As a result, synthetic methods enabling their rapid construction and direct structural modification are of significant value. The Minisci reaction, which allows direct C–H functionalization of heteroarenes through a radical pathway, is a powerful method for the synthesis of multiply substituted heterocycles[4–10]. The reaction involves the addition of carbon-centered radicals to heteroarenes, typically pyridine-based heteroarenes, followed by formal loss of a hydrogen atom. The Minisci reaction is important, and tremendous efforts and advances have been made in this area in the past decades[4]. However, there are still many challenges in terms of reactivity and regioselectivity (Fig. 1a). The major limitation to the Minisci reaction is the lack of regioselectivity which usually leads to the formation of regioisomeric mixtures, moderate chemical yields and the need for product purification[4–6]. In order to facilitate the radical addition to chemically inert heteroarenes, strong acid is typically used as a stoichiometric additive to protonate the basic heteroarenes and thus lower the energy of their LUMO[6]. Carbon-centered alkyl radicals are often generated from radical precursors —typically carboxylic acid derivatives—with transition metals and oxidants, typically a combination of $Ag(I)/S_2O_4^{2-}$ or $Fe(II)/H_2O_2$ at elevated temperatures. A modified method which can address the aforementioned challenges to the Minisci reaction is always in demand. Such a method ideally should control regioselectivity, avoid the use of precious metals and oxidants and proceed under nonacidic conditions at room temperature.

Nitrogen-centered radicals (NCRs) are a class of valuable synthetic intermediates, and have become the focus of significant research efforts in recent years[11–14]. Radical translocation processes triggered by NCRs, such as 1,5-hydrogen atom transfers (1,5-HAT), demonstrated by the well-established Hofmann–Löffler–Freytag (HLF) reaction, provide an attractive approach for the controllable and selective functionalization of remote inert $C(sp^3)$–H bonds[15–19]. Recently, the photoredox catalysis[20,21] combined with classic HAT[22] provides an alternative synthetic tool for remote $C(sp^3)$–H functionalization. This strategy offers a marvelous pathway to selectively achieve mild C–H bond cleavage and C–X (X = halides) and C–N bond formation[19]. Furthermore, interrupted HLF reactions using electron-deficient alkenes[23–27], vinyl boronic acids[28,29], allyl sulfones[30], allylic chlorides[31], etc.[32,33] as carbon-centered radical traps open a new window for remote $C(sp^3)$–C bond formation.

Despite these advances, an interrupted HLF reaction by trapping of the carbon-centered radicals with non-prefunctionalized heteroarenes leading to remote $C(sp^3)$–H (hetero)arylation under photoredox catalysis is still challenging and remains largely unexplored. Remote $C(sp^3)$–H (hetero)arylation in the assistance of oxygen-centered radical-triggered 1,5-HAT processes has been achieved by several groups[34–36]. Recently, Zhu et al.[37] and Nagib et al.[38] independently developed a Cu-catalyzed (hetero)arylation of remote $C(sp^3)$–H bonds with boronic acids as the cross-coupling partners. More recently, Zhu et al.[39] reported a hypervalent iodine-promoted remote $C(sp^3)$–H heteroarylation of amides. Inspired by these reports on remote $C(sp^3)$–H (hetero)arylation, here, we report an amidyl radical-triggered, transition metal free and site-selective remote $C(sp^3)$–H heteroarylation with non-prefunctionalized heteroarenes under photoredox conditions (Fig. 1b).

## Results

**Reaction optimization.** At the outset of this project, the purine derivative (**2a**) was chosen to couple with the amidyl radical precursor (**1a**). Purine derivatives are bases found in DNA and RNA, and they also possess antiviral activity[40–42]. There are three types of $C(sp^2)$–H bonds, at C2, C6, and C8 in purines, and they are the cause of regioselectivity issues[43]. After much effort, conditions for the coupling of the hydroxamide (**1a**) with 9-benzyl-9*H*-purine (**2a**) were successfully established. Details of the optimization of the reaction conditions are in the Supplementary Tables 1–6. A mixture of **1a** (0.2 mmol, 1 equiv) and **2a** (2.5 equiv) in DMSO was irradiated with 90 W blue LEDs in the presence of 2 mol% of (2,4,6-tri(9H-carbazol-9-yl)-5-chloroiso-phthalonitrile (3CzClIPN) as the photocatalyst, and $K_2CO_3$ as the base at room temperature. The C6 alkylated purine (**3**) was isolated in 89% yield as a single regioisomer (Table 1, entry 1). The conditions are weakly basic, and no transition metal and external oxidant are necessary. The reaction is highly site-selective, and no other isomers were observed. The photocatalyst 3CzClIPN synthesized by Zeitler[44] has proved to have remarkable synthetic potential in this reaction. Control experiments showed that light, base, and a nitrogen atmosphere are critical to this reaction (Table 1, entries 2–4). When the reaction proceeds without the photocatalyst, product **3** can be obtained in 65% isolated yield (Table 1, entry 5). It was speculated that the photocatalyst-free reaction is enabled by the formation of an electron–donor–acceptor (EDA) complex[45,46]. However, it was found that the yields of photocatalyst-free reactions were very sensitive to substrates (vide infra), and thus the

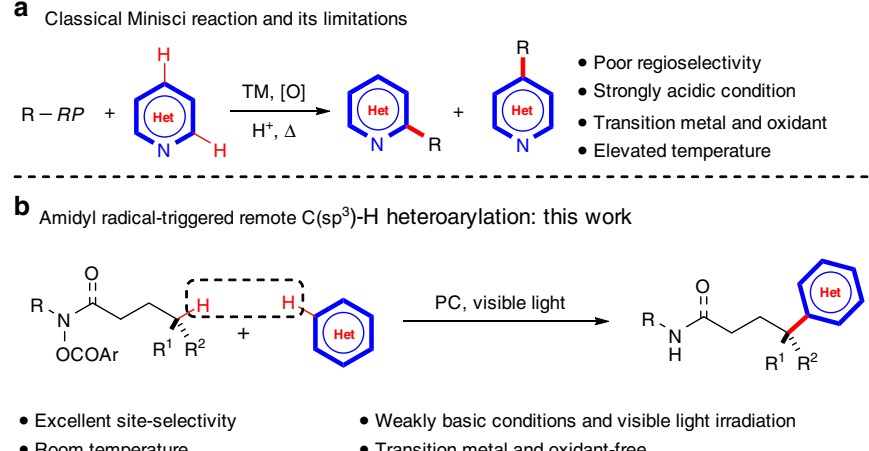

**a** Classical Minisci reaction and its limitations

- Poor regioselectivity
- Strongly acidic condition
- Transition metal and oxidant
- Elevated temperature

**b** Amidyl radical-triggered remote $C(sp^3)$-H heteroarylation: this work

PC, visible light

- Excellent site-selectivity
- Room temperature
- Weakly basic conditions and visible light irradiation
- Transition metal and oxidant-free

**Fig. 1** Alkylation of hereroarenes. **a** Classical Minisci reaction and its limitations. **b** Amidyl radical-triggered remote $C(sp^3)$-H heteroarylation: this work. RP radical precursor

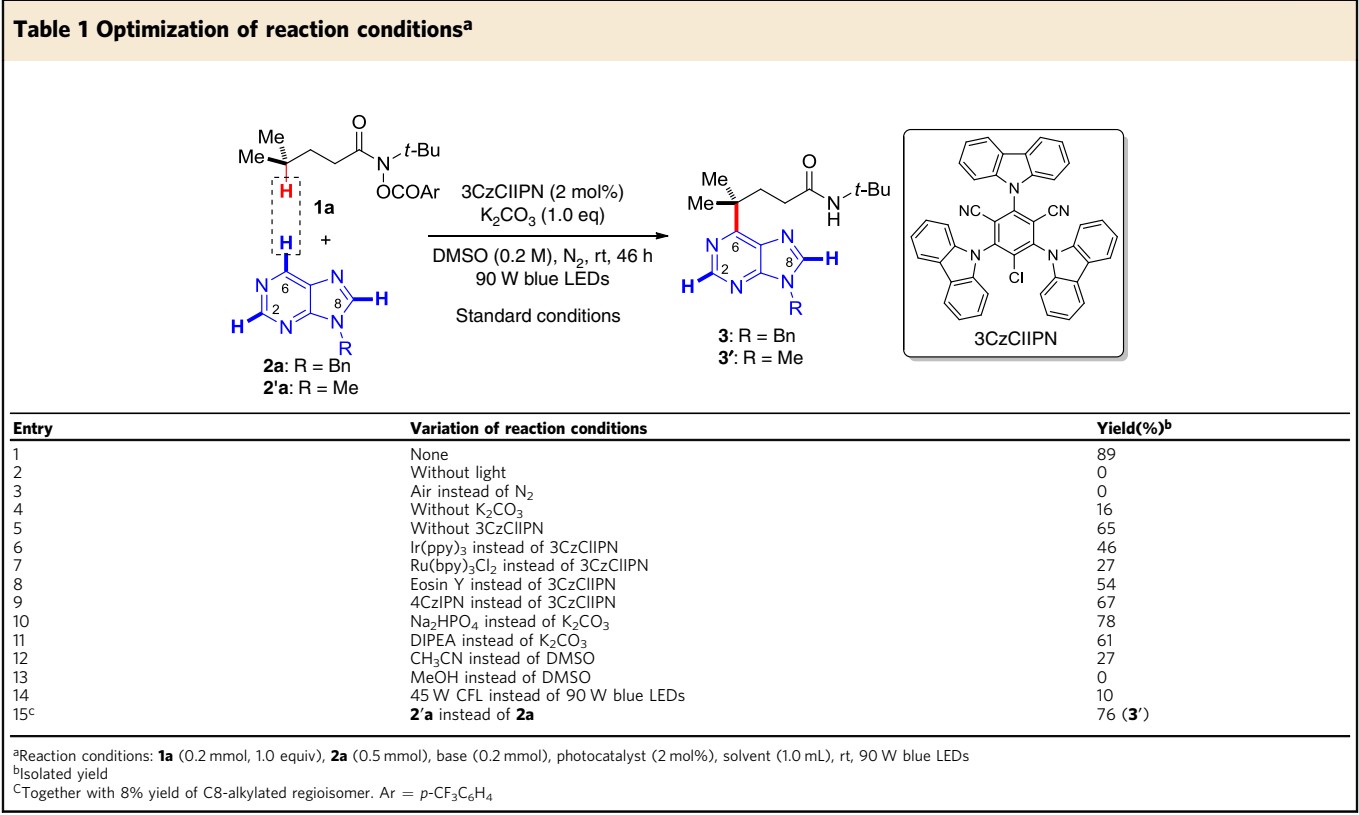

**Table 1 Optimization of reaction conditions[a]**

| Entry | Variation of reaction conditions | Yield(%)[b] |
|---|---|---|
| 1 | None | 89 |
| 2 | Without light | 0 |
| 3 | Air instead of N₂ | 0 |
| 4 | Without K₂CO₃ | 16 |
| 5 | Without 3CzClIPN | 65 |
| 6 | Ir(ppy)₃ instead of 3CzClIPN | 46 |
| 7 | Ru(bpy)₃Cl₂ instead of 3CzClIPN | 27 |
| 8 | Eosin Y instead of 3CzClIPN | 54 |
| 9 | 4CzIPN instead of 3CzClIPN | 67 |
| 10 | Na₂HPO₄ instead of K₂CO₃ | 78 |
| 11 | DIPEA instead of K₂CO₃ | 61 |
| 12 | CH₃CN instead of DMSO | 27 |
| 13 | MeOH instead of DMSO | 0 |
| 14 | 45 W CFL instead of 90 W blue LEDs | 10 |
| 15[c] | 2′a instead of 2a | 76 (3′) |

[a]Reaction conditions: **1a** (0.2 mmol, 1.0 equiv), **2a** (0.5 mmol), base (0.2 mmol), photocatalyst (2 mol%), solvent (1.0 mL), rt, 90 W blue LEDs
[b]Isolated yield
[c]Together with 8% yield of C8-alkylated regioisomer. Ar = p-CF₃C₆H₄

76% yield together with 8% yield of C8-alkylated regioisomer (Table 1, entry 15).

**Mechanistic studies**. A proposed mechanism for this site-selective remote C(sp³)–H heteroarylation is outlined in Fig. 2. Upon single-electron reduction by the excited photocatalyst 3CzClIPN*, the hydroxamide (**1a**), forms an amidyl radical (**A**), a carboxylate anion (ArCO₂⁻), and 3CzClIPN⁺•. A 1,5-HAT from carbon to nitrogen provides a carbon-centered radical (**B**) which is trapped by **2a**, giving the radical intermediate (**C**). This radical (**C**) undergoes formal hydrogen atom loss through a stepwise electron transfer (ET)/proton transfer (PT)[48–51] with the assistance of the base and 3CzClIPN⁺•, delivering the final product (**3**) and regenerating the photocatalyst 3CzClIPN.

To better understand and validate this mechanistic hypothesis, density functional theory (DFT) calculations using the M06–2X functional[52] were performed in an investigation of the energetics of the proposed mechanism (Fig. 3a). First, the hydroxamide (**1a**) is reduced by the excited photocatalyst to form an amidyl radical (**A**) via a SET and with an activation free energy of 3.3 kcal mol⁻¹, estimated according to the Marcus theory[53–55]. The SET step is slightly endergonic (1.4 kcal mol⁻¹). Subsequently, a 1,5-HAT proceeds via **TS1**, generating a carbon-centered radical (**B**). This step is exergonic by 14.1 kcal mol⁻¹ and has an 8.4 kcal mol⁻¹ energy barrier, suggesting that the formation at room temperature of the radical (**B**) is feasible. The radical intermediate (**B**) then can attack any of three different carbons C2, C6, or C8 of **2a** to activate different C(sp²)–H bonds and giving the additional intermediate (**C**). The C6–H bond activation in **2a** via **TS2** only requires it to overcome an 11.7 kcal mol⁻¹ barrier, which is lower than the bond activation associated with C2–H (14.7 kcal mol⁻¹, **TS2'** with respect to **B**) or C8–H (16.2 kcal mol⁻¹, **TS2"** with respect to **B**). With 15.1 kcal mol⁻¹, 11.0 kcal mol⁻¹, 13.5 kcal mol⁻¹ for C6–H, C2–H bond and C8–H bond, respectively, and this is calculated to be the most exergonic of the three C–H bond activation

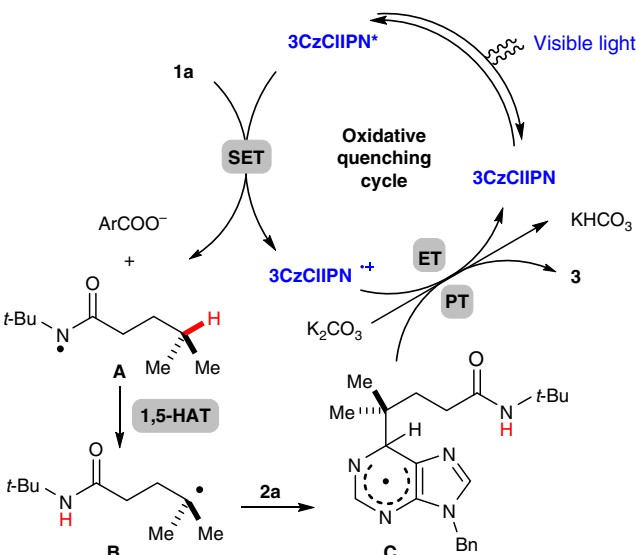

**Fig. 2** Proposed mechanism. The proposed mechanism for the interrupted HLF reaction with non-prefunctionalized heteroarenes

photocatalyst was used in all reactions. Replacing 3CzClIPN with other photocatalysts, such as Ir(ppy)₃, Ru(bpy)₃Cl₂, Eosin Y, and 1,2,3,5-tetrakis(carbazol-9-yl)-4,6-dicyanobenzene (4CzIPN)[47], resulted in lower yields of **3** (Table 1, entries 6–9). Other bases (Na₂HPO₄ and DIPEA) and solvents (CH₃CN and MeOH) were also investigated, but none gave improved results (Table 1, entries 10–13). In addition, much inferior yield was obtained (10%) if the reaction was irradiation under compact fluorescent light (CFL) (Table 1, entry 14). When 9-methyl-9H-purine (**2′a**) was employed instead of **2a**, the C6-alkylated purine (**3′**) was isolated in

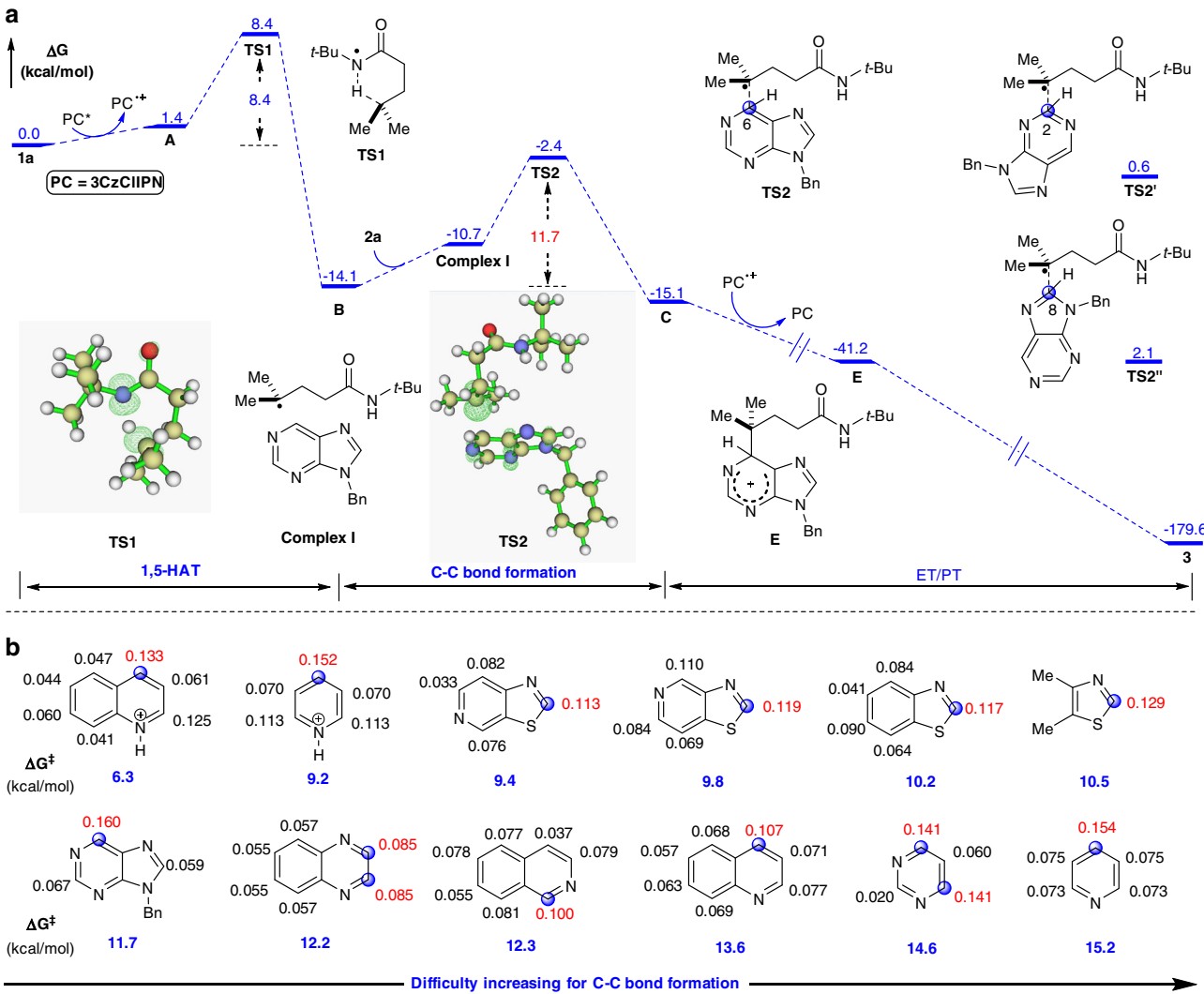

**Fig. 3** Density functional theory calculations. **a** Computed Gibbs free energy profile for the redox-neutral coupling reaction of **1a** and **2a** catalyzed by photocatalyst 3CzClIPN and spin density of transition states **TS1** and **TS2**. Energies are given in kcal mol$^{-1}$. **b** Fukui indices and free energy barriers for the C–C formation steps at each position predicted by Fukui indices ($f^+$) (Highlighted in blue balls)

processes. This suggests that the C6–H bond activation is kinetically and thermodynamically more favorable than either C2–H or C8–H bond activation. The findings are consistent with our experimental observation that only the C6-alkylated purine (**3**) was isolated. The calculated results also show that the C(sp$^2$)–H site-selectivity in the redox-neutral coupling reaction is determined in this additional step, which is identified as the rate-determining step of the reaction. The overall energy requirement of the reaction, generally known as "the overall energy barrier"[56] is calculated to be 11.7 kcal mol$^{-1}$. Finally, the radical **C** proceeds by oxidation and deprotonation to deliver the product **3** and regenerate the photocatalyst 3CzClIPN (for details, see Supplementary Fig. 7). The entire C6-alkylated coupling pathway is exergonic by 179.6 kcal mol$^{-1}$, and has an overall energy barrier of 11.7 kcal mol$^{-1}$.

The Fukui function is widely used in prediction of reactive sites[57,58]. The atom with the maximal value of Fukui index is predicted to be the preferred reactive site. The Fukui index of C6 in the reactant (**2a**) is calculated with multiwfn 3.6[59] to be 0.160, larger than that at C2 (0.067) or C8 (0.059), indicating that C6 is the most favorable reactive site (Fig. 3b), which is consistent with it being the experimentally observed reactive site. Motivated by the exclusive C(sp$^2$) site-selectivity in both experiment and

theory, we set to calculated Fukui indices of a variety of heteroarenes, as well as the free energy barriers in the rate-determining addition step for C–C bond formation. This should allow prediction of the site-selectivity and reactivity of these compounds. As shown in Fig. 3b, the benzothiazole, thiazole, and thiazolopyridines are predicted to be more reactive than purine due to the lower C–C formation energy barriers while quinoxaline, quinoline, isoquinoline, pyrimidine, and pyridine are calculated to be less reactive than purine due to higher C–C formation energy barriers. Carbon atoms with more positive Fukui indices are predicted at the positions where the C–C formation takes place. Protonated pyridine and quinoline are more reactive compounds with much lower energy barriers (9.2 and 6.3 kcal mol$^{-1}$, respectively), which is consistent with experimental observations in the classical acid-promoted Minisci reaction[5].

**Substrate scope.** To substantiate our predictions and demonstrate the generality and practicality of this remote C(sp$^3$)–H heteroarylation, we investigated the substrate scope of the heteroarenes under the optimized conditions (Fig. 4a). Reactions of purine analogues afforded products **3** and **4** selectively alkylated at the C6 position in 89 and 95% yields, respectively. When

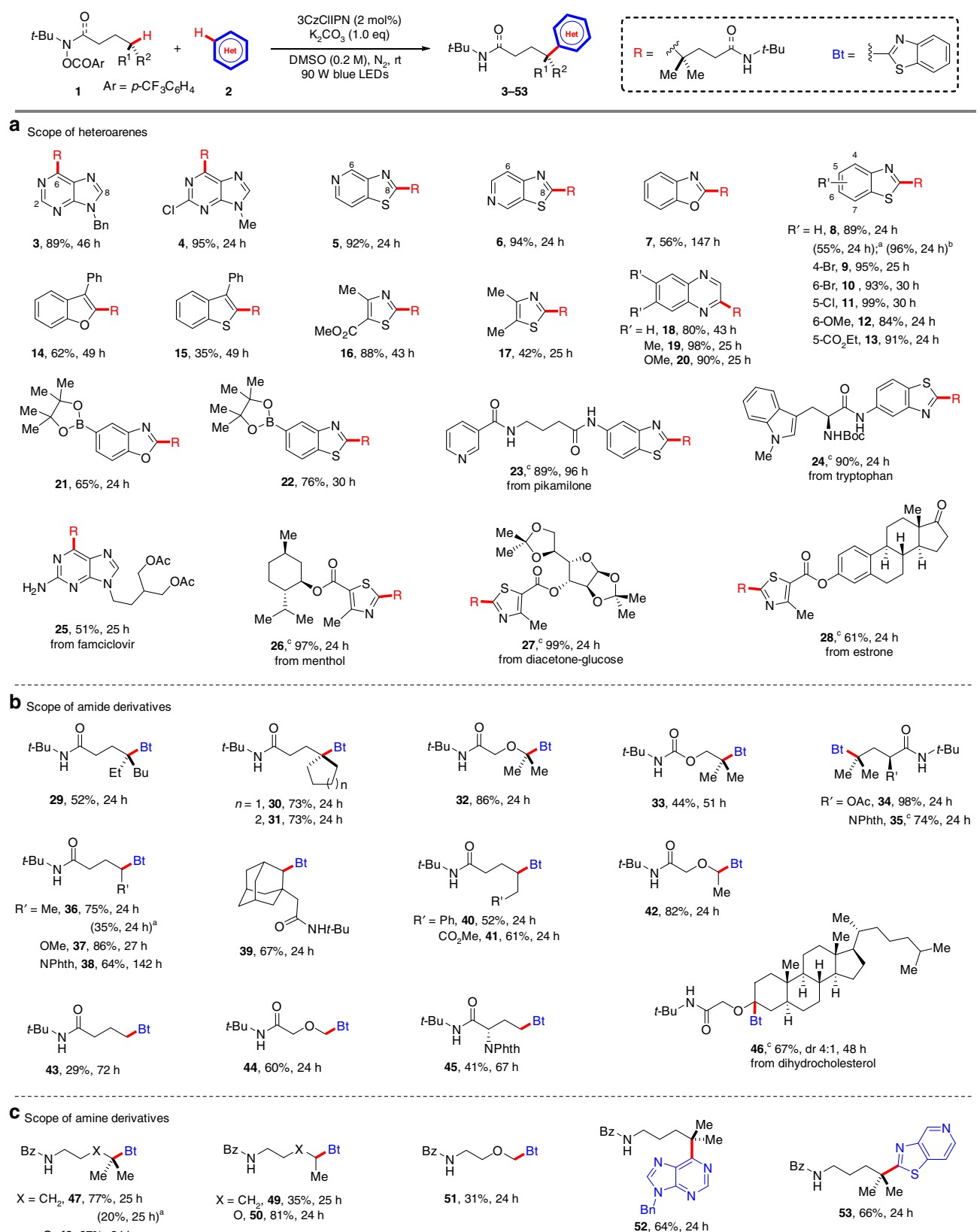

**Fig. 4** Scope of the substrates. **a** Scope of heteroarenes. **b** Scope of amide derivatives. **c** Scope of amine derivatives. Reaction conditions: **1** (0.2 mmol, 1.0 equiv), **2** (0.5 mmol), K$_2$CO$_3$ (0.2 mmol), and 3CzClIPN (2 mol%) in DMSO (1.0 mL), rt, 90 W blue LEDs. Yields are based on the isolated products. [a]Without 3CzClIPN. [b]5 mmol scale of **1a**. [c]0.1 mmol scale of **1a**, **1h**, and **1s**, respectively

**a** pH-controlled orthogonal C–H functionalization of heteroarenes

*C–H alkylation of benzothiazole*

*C–H alkylation of quinoline*

**b** Radical polarity-controlled orthogonal C–H functionalization of heteroarenes

*C–H alkylation of benzothiazole*

*C–H amidation of indole*

*C–H amidation of indole*

*C–H alkylation of benzothiazole*

**Fig. 5** Sequential and orthogonal C–H functionalization of heteroarenes. **a** pH-controlled orthogonal C–H functionalization of heteroarenes. **b** Radical polarity-controlled orthogonal C–H functionalization of heteroarenes. The structure of the R group is shown in Fig. 4

thiazolopyridines were used, the sole C8-alkylated products (**5** and **6**) were obtained with 92 and 94% yields, respectively. Benzoxazole was also proven to be an effective substrate, producing its alkylated analogue (**7**) in 56% yield. Reactions of benzothiazole and its derivatives with electron-donating or electron-withdrawing substituents proceeded smoothly with excellent regioselectivity to afford the desired products (**8–13**) with 84–99% yields. Other heteroarenes, such as benzothiophene, benzofuran, thiazoles, and quinoxalines, were all effective substrates, affording a variety of alkylated heteroarenes (**14–20**) in 35–98% yields. All predicted reactive heteroarenes were efficient substrates in this reaction, and new C−C bonds were formed at the positions indicated by the Fukui function. As predicted, less reactive heteroarenes, such as quinoline, isoquinoline, pyrimidine, and pyridine, failed to undergo this transformation (other unreactive heteroarenes are listed in the Supplementary Fig. 5). Boronates were tolerated in this protocol, and boronate-derived benzoxazole and benzothiazole could be alkylated without involving the boronates to give the products **21** and **22** in 65 and 76% yields, respectively. Significantly, the method can be applied to late-stage modification of biologically relevant molecules. For example, the reaction of a pikamilone derivative gave the desired product (**23**) in 89% yield. However, the pyridine moiety embedded in pikamilone, which was easily alkylated in the classical Minisci reaction, remained intact in the reaction. The electron-rich indole moiety could survive these alkylation conditions, producing **24** in 90% yield. Famciclovir, a guanosine-containing antiviral drug used in the treatment of various herpes viral infections[60], could be alkylated to give the product (**25**) in 51% yield and benzothiazoles connected with biologically important molecules, such as menthol, glucose, and estrone, effectively afforded the corresponding alkylation products (**26–28**) in 61–99% yields.

The scope of hydroxamides was then explored using benzothiazole (**2 f**) as the coupling partner (Fig. 4b). Hydroxamides with tertiary C(sp$^3$)−H bonds were well tolerated, giving

products **29−35** in 44–98% yields. Secondary C(sp$^3$)−H bonds could also be heteroarylated, affording products **36−42** in 52 −86% yields. It was found that heteroarylation of primary C(sp$^3$) −H bonds were susceptible, giving **43−45** albeit with moderate 29−60% yields. Late-stage functionalization of natural product derivatives was examined, as demonstrated by the steroidal derivative (**1 s**), which gave product **46** in 67% yield. Compounds with various functional groups, such as amides (**35** and **45**), ethers (**32**, **37**, **42** and **44**), and esters (**33−34** and **41**), are all amenable to this reaction. Amino acid derivatives also reacted, giving **35** and **45** in satisfactory yields. In addition, several benzamide derivatives were heteroarylated successfully (31−87% yields for **47−53**) (Fig. 4c).

Compounds **8**, **36**, and **47** could be also prepared under photocatalyst-free conditions, but with significantly lower yields (**8**: 55 vs 89%; **36**: 35 vs 75%; **47**: 20 vs 77%). The reaction could be easily completed at the gram scale (5 mmol), as demonstrated by the synthesis of **8**, which was isolated in 96% yield.

Our conditions are weakly basic, which is orthogonal to the acidic conditions of the classical Minisci reaction. By taking advantage of these pH-dependent conditions, a sequential C−H functionalization of different heteroarenes in the same molecule could be achieved. As demonstrated in Fig. 5a, heteroarene **2a'**, with benzothiazole and quinoline moieties, coupled with **1a** under our standard conditions to give only the benzothiazole-alkylated product (**54**), which was isolated in 81% yield. The quinoline moiety remained unaffected due to its inertness under basic conditions. As shown in Fig. 3b, quinoline can be activated by protonation, and the C2 position of quinoline **54** could be further alkylated with the redox-active ester (**55**) under the acidic conditions established by Fu et al[61]. to give the dialkylation product (**56**) in 85% yield. The indole-tethered benzothiazole (**2b'**) could be also functionalized sequentially and controlled orthogonally by radical polarity (Fig. 5b). The electron-poor benzothiazole moiety was alkylated with a nucleophilic alkyl radical under these conditions, giving an 80% yield. Subsequently,

**Fig. 6** Competition between different 1,5-HAT processes and subsequent C–C formation. Ar = $p$-CF$_3$C$_6$H$_4$

the electron-rich indole motif was amidated in 99% yield with an electrophilic amidyl radical under our previously developed amidation conditions[62]. Moreover, the heteroarene (**2b'**) could also be amidated in 99% yield while the benzothiazole moiety remained intact, and was subsequent alkylated under our standard conditions, furnishing product **59** in 51% yield.

Since both *N*-alkyl and the acyl sides of amidyl radical can undergo a 1,5-HAT process, we were interested in the competition between different 1,5-HAT processes. The hydroxamide derivative (**1 y**), which presents two similarly inert tertiary C−H bonds, both of which could go through 1,5-HAT, was coupled with benzothiazole (**2 f**) under the standard conditions (Fig. 6). The result revealed that a HAT occurred at both tertiary C−H bonds, delivering products **61** and **62** in 55 and 27% yields, respectively. Their structures were determined by careful 2D NMR (HSQC and HMBC) experiments. Heteroarylation at the *N*-alkyl side, as in **61**, was easier than at the acyl side (**62**). This could be explained by the DFT-computed free energy profile for the different processes (for details, see Supplementary Fig. 9). It was found that the generation of **61** (via **TS6**, the energy barrier is 12.8 kcal mol$^{-1}$) was favored over the process that would be involved in the generation of **62** (via **TS7**, the energy barrier is 14.0 kcal mol$^{-1}$).

## Discussion

In summary, we have developed an amidyl radical-triggered Minisci-type reaction that allows site-selective C(sp$^3$)–H/C(sp$^2$)–H coupling under photoredox conditions. Non-prefunctionalized heteroarenes are the coupling partners, and this approach provides a mild and highly regioselective way to afford the remote C (sp$^3$)–H heteroarylated amides at room temperature under conditions that are transition metal free, weakly basic, and redox neutral. The reaction features good functional group tolerance and broad substrate scope, and provides a unique strategy for the late-stage functionalization of complex bioactive molecules. Sequential and orthogonal C–H functionalization of different heteroarenes by taking advantage pH value or polarity of radicals has also been achieved. DFT calculations explain and can predict the site-selectivity and reactivity of this reaction. The use of this strategy

will permit further discovery of inert C(sp$^3$)−H functionalization, and is currently in progress in our laboratory.

## Methods

**General procedure**. Hydroxamide (**1**) (0.2 mmol, 1.0 equiv), heteroarene (**2**) (0.5 mmol, 2.5 equiv), K$_2$CO$_3$ (0.2 mmol, 1.0 equiv), and 3CzClIPN (2 mol%) were added successively to an 8 -mL vial. The vial was evacuated and backfilled with N$_2$ 3–5 times. DMSO (1.0 mL, 0.2 M) was added under N$_2$ with a syringe. The mixture was then irradiated by two 90 W blue LEDs. After the reaction was complete as judged by TLC analysis, the mixture was quenched by adding 20 mL of saturated aqueous NaCl and 20 mL of EtOAc. The layers were separated and the aqueous layer was extracted with EtOAc (2 × 20 mL). The combined organic layers were washed with saturated NaHCO$_3$ (20 mL) and brine (20 mL), successively, and then evaporated. The crude product was purified by column chromatography on silica gel to afford the desired product.

## Data availability

The authors declare that the main data supporting the findings of this study, including experimental procedures and compound characterization, are available within the article and its Supplementary Information files, or from the corresponding author upon reasonable request.

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

## Acknowledgements

Financial support from National Natural Science Foundation of China (21672098, 21732003, and 21703118), National Key Research and Development Program of China (2018YFC0310900), and Shandong Provincial Natural Science Foundation, China (ZR2017MB038) is acknowledged. We are also grateful to the High Performance Computing Center of Qufu Normal University for performing the quantum chemical calculations.

## Author contributions

S.Y. designed and guided this project. H.C. and S.Y. are responsible for the plan and implementation of the experimental work. W.F. and X.-A.Y. are responsible for the calculation studies. All authors co-wrote the paper, analyzed the data, discussed the results, and commented on the paper.

## Competing interests

The authors declare no competing interests.
