## [Peer Review File · Nature Communications]

Reviewer #1 (Remarks to the Author):

In this manuscript, Yu and coworkers describe the development of a method for the delta C-H heteroarylation of various amides. The strategy entails a 1,5-HAT from a nitrogen-centered radical followed by trapping of the delta carbon radical with heteroarenes by a Minisci mechanism. The amidyl radicals are accessed by SET reduction of a preinstalled hydroxyamide N-O bond by an organic photocatalyst. The scope of heteroarenes used is very broad, including many 5- and 6-membered N-containing heterocycles. The amide scope is also well-developed with various types of C-H bonds as well as backbones that are tolerated. The regioselectivity is high as expected for the delta C-H, and perhaps more surprising for the heteroarenes. Interesting competition experiments are also included to demonstrate the synthetic utility of some of these observed selectivities. DFT calculations are provided as mechanistic support for the regioselective addition into heteroarenes (Fukui indices) as well as the competition between delta C-H's on either side of the amide. A reasonable energy diagram is also provided. Overall, the method is well-developed and evaluated, and this strategy offers a nice complement to current methods for remote C-H heteroarylation. However, the introduction is poorly written and several key references to similar work are omitted (especially Chem. Sci., 2019, 10, 6915). Therefore, publication is recommended only after major revisions, as indicated below.

Major Revisions

1. The title, abstract, and introduction do not adequately describe the novelty or innovation of the new reactions developed here. For example, the title is much too vague. As written, it would refer to an entire field of chemistry. Instead, the title should refer only to the key addressed reaction which is the delta C-H heteroarylation of amides.
2. Similarly, the abstract and intro provide detailed discussion of the Minisci reaction, making it seem that this paper has made innovations in this area, when it has not. Instead, the Minisci aspect is merely the radical trap at the remote carbon radical. The most important aspect of this paper is that it interrupts an HLF mechanism with a Minisci reaction. Since this is the novelty, the emphasis should have been on the interrupted HLF, not the Minisci. The abstract also confusingly refers to regioselectivity issues (unclear which ones) in the Minisci reaction, which is pretty robust for alpha-selectivity. Perhaps the authors are referring to the position on the heteroarenes. If so, it is unclear how or whether the authors have addressed this, or merely used heteroarenes that are innately selective.

3. Most importantly, the introduction does not adequately describe the state of the art in the field, and is misleading as written. For example, there have been several relevant delta-C-H heteroarylations by 1,5-HAT reported recently.

Three mediated by O-centered radicals include:

Chen/He: Chem. Sci., 2019,10, 688 (Ref 17)

Baik/Hong: Angew. Chem. Int. Ed. 2018, 57, 15517

Zhu: Nat. Commun., 2018, 9, 3343.

Three mediated by N-centered radicals, including with boronic acid traps (Ref 35-36) as well as a Minisci trap (most relevant):

Zhu: Chem. Sci., 2019, 10, 6915

While these previous advances do not preclude publication of this nice new work. It will be important for the authors to put their innovations into the correct context. For example, the O-centered radicals are much more electrophilic, so it is not surprising that their reactivity has been much better developed over several decades, including the recent Minisci trapping reactions. However, the switch to an interrupted HLF mechanism is challenging, and has not been accomplished until recently. Among the 3 reports, 2 use N-F as a radical precursor (Refs 35-36) and the 2019 Zhu work uses hypervalent iodine to access N-centered radicals. In this manuscript, by using an organophotocatalyst to break an N-O bond, the authors have enabled some nice divergent reactivity and selectivity. However, without an explicit introduction to the previous work, this novelty can not be appreciated fully.

4. References: In general, many key references are buried in sentences that do not accurately describe the innovation. Some examples include:

Page 2, line 31 - refers to the Minisci reaction, when it should be a Minisci trap of an intramolecular HAT (Ref 17)

Page 2, line 39 - comparison to conditions for alpha-C-H abstraction in the Minisci reaction are irrelevant, since this work entails HAT from N-centered radicals. Only generation of N-centered radicals would be relevant.

Page 3, top - this should be the key intro about N-centered radicals and the HLF reaction. Also, the difference between Refs 37-48 should be described. Most are interrupted HLF reaction. Ref 44 and 45 are the same. Ref 46 is confusingly the only O-radical HAT, but is not one of the delta arylations

mentioned above. If this were the intro paragraph, it would be the correct place to explain the contributions above.

5. Regioselectivity: Page 4, first paragraph refers to "intractable regioselectivity issues." This should include a reference. It is also strange because there is no mention of regioselectivity in the optimization, or later on. Does this mean it was >20:1? If so, and given the DFT support, then what are the "intractable regioselectivity issues"?

6. PCET: Page 6 refers to PCET, but there is no evidence that the proton and electron are coupled in this mechanism. It is just the net loss of an H-atom. It could simply be an asynchronous oxidation and deprotonation. The references cited go to great lengths to determine if those events are coupled, but this paper does not. A simple loss of H-atom would do.

7. References: The references to seminal work on Marcus Theory and the Fukui Function are a little strange. It would be more helpful to cite recent applications of these tools to C-H functionalization, rather than include definition references that could be found in a textbook. Also, as currently written, the Fukui Index explanation (line 129) is incorrect. The Fukui Index does not "favor" reactivity. Instead, it 'may explain' selectivity. In this manuscript, the authors have correlated their observed reactivity with Fukui values that explain electron density perturbations. This is interesting, but should be stated properly.

8. The third part of Figure 3 is labeled as PCET. It should just be HAT. Otherwise, there should be DFT support for PCET.

9. Throughout the manuscript and SI, the hydroxamides are referred to as hydroxyamines, but they are amides.

10. Competition: Line 221 refers to "both amide and amine moieties" but both are amidyl radicals. Perhaps reword to the N-alkyl side and the Acyl side?

11. Line 230 states both "kinetically and thermodynamically favored." Which is it? Also, a 3 kcal/mol difference is shown, but only 2:1 selectivity is observed.

Reviewer #2 (Remarks to the Author):

In this manuscript Yu and coworkers disclosed a Minisci-type alkylation reaction of N-heteroarenes via amidyl radical mediated 1,5-HAT process under organic photoredox catalysis. Different from previous Minisci-type alkylation reactions that requiring acidic conditions, this reaction proceeded well under basic conditions and provided an alternative strategy for the acid sensitive substrates. A broad scope of both hydroxylamines and N-heteroarenes was reported, and functionalization complex bioactive molecules were also demonstrated. The reactivity and regioselectivity were investigated by DFT calculations and the results were consistent with the experimental observation. Overall, this paper well-written and publication in Nature Communication is suggested after addressing some minor issues:

1. Can the authors speculate on the reason why this Minisci-type alkylation reaction proceeds well under basic conditions?
2. What if this reaction is conducted in the acidic conditions using the same starting material and organic photoredox catalysis? Particularly for the unreactive substrates under basic conditions, such as quinoline and pyridine.
3. Recent reports on Minisci-type alkylation via 1,5-HAT process should be cited:

Nat. Commun. 2018, 9, 3343; Chem. Sci. 2019, 6915.

Reviewer #3 (Remarks to the Author):

The present manuscript describes a novel and facile approach to enabling remote Minisci-type heteroarylations using N-centered amidyl radicals with an interesting mechanism for radical generation under metal-free photo-redox conditions. The novelty of this transformation is that it is a unique example with excellent regioselectivity compared to the well-known Minisci-type alkylation of heterocycles. More than 50 examples and 8 classes of heterocycles (including purines, thiazolopyridines, benzoxazole, benzothiazole, benzothiophenes, benzofurans, thiazoles and quinoxalines) were amenable for this alkylation transformation. Furthermore, a cascade direct C-H bond functionalization of different heteroarenes by taking advantage pH value or polarity of radicals has also been studied in the manuscript. Finally, the investigation of mechanism in the paper is well conducted, and an appropriate radical mechanistic proposal is given based on experimental results,

and the site-selectivity and reactivity of this reaction was also well-supported by DFT calculations. Therefore, I suggest this manuscript to be published on Nature Commun.

1. Authors should include at least one sentence in the introduction section of the manuscript to illustrate the importance of visible light in an organic reaction.
2. the base K_2CO_3 seems quite important for this transformation, the authors should give some comments on it.
3. When the reaction was irradiation under compact fluorescent light (CFL), is it possible to obtain the product of 3?

Our point-to-point responses to the comments from the reviewers have been highlighted as followed:

Reviewer 1:

Q: In this manuscript, Yu and coworkers describe the development of a method for the delta C-H heteroarylation of various amides. The strategy entails a 1,5-HAT from a nitrogen-centered radical followed by trapping of the delta carbon radical with heteroarenes by a Minisci mechanism. The amidyl radicals are accessed by SET reduction of a preinstalled hydroxyamide N-O bond by an organic photocatalyst. The scope of heteroarenes used is very broad, including many 5- and 6-membered N-containing heterocycles. The amide scope is also well-developed with various types of C-H bonds as well as backbones that are tolerated. The regioselectivity is high as expected for the delta C-H, and perhaps more surprising for the heteroarenes. Interesting competition experiments are also included to demonstrate the synthetic utility of some of these observed selectivities. DFT calculations are provided as mechanistic support for the regioselective addition into heteroarenes (Fukui indices) as well as the competition between delta C-H's on either side of the amide. A reasonable energy diagram is also provided. Overall, the method is well-developed and evaluated, and this strategy offers a nice complement to current methods for remote C-H heteroarylation. However, the introduction is poorly written and several key references to similar work are omitted (especially Chem. Sci., 2019, 10, 6915). Therefore, publication is recommended only after major revisions, as indicated below.

A: We thanks for these insightful comments.

Q: 1. The title, abstract, and introduction do not adequately describe the novelty or innovation of the new reactions developed here. For example, the title is much too vague. As written, it would refer to an entire field of chemistry. Instead, the title should refer only to the key addressed reaction which is the delta C-H heteroarylation of amides.

A: We accept this suggestion and a more specific title is adopted. Since the gamma C-H of carbonyls (acyl side, Figures 4a and 4b) and delta C-H of amine derivatives (N-alkyl side, Figure 4c) are heteroarylated, the title is changed to "Site-Selective Remote C(sp³)-H Heteroarylation of Amides via Organic Photoredox Catalysis".

Q: 2. Similarly, the abstract and intro provide detailed discussion of the Minisci reaction, making it

seem that this paper has made innovations in this area, when it has not. Instead, the Minisci aspect is merely the radical trap at the remote carbon radical. The most important aspect of this paper is that it interrupts an HLF mechanism with a Minisci reaction. Since this is the novelty, the emphasis should have been on the interrupted HLF, not the Minisci. The abstract also confusingly refers to regioselectivity issues (unclear which ones) in the Minisci reaction, which is pretty robust for alpha-selectivity. Perhaps the authors are referring to the position on the heteroarenes. If so, it is unclear how or whether the authors have addressed this, or merely used heteroarenes that are innately selective.

A: We thanks for these insightful comments and suggestions. The abstract and introduction are reorganized, mainly focused on interrupted HLF reactions (See the revised manuscript for details. The changes are highlighted in yellow.) The introduction to Minisci reaction is weakened. Regioselectivity issue in the abstract is removed. The revised abstract is quoted as the following:

“An amidyl radical-triggered site-selective remote C(sp³)–H heteroarylation of amides under organic photoredox conditions is reported. This approach provides a mild and highly regioselective reaction affording remote C(sp³)–H heteroarylated amides at room temperature under transition-metal free, weakly basic, and redox-neutral conditions. Non-prefunctionalized heteroarenes, such as purines, thiazolopyridines, benzoxazole, benzothiazoles, benzothiophene, benzofuran, thiazoles and quinoxalines, can be alkylated directly. Sequential and orthogonal C–H functionalization of different heteroarenes by taking advantage pH value or polarity of radicals has also been achieved. DFT calculations explain and can predict the site-selectivity and reactivity of this reaction. This strategy expands the scope of the Minisci reaction and serves as its alternative and potential complement.”

Q: 3. Most importantly, the introduction does not adequately describe the state of the art in the field, and is misleading as written. For example, there have been several relevant delta-C-H heteroarylations by 1,5-HAT reported recently.

Three mediated by O-centered radicals include:

Chen/He: Chem. Sci., 2019,10, 688 (Ref 17)

Baik/Hong: Angew. Chem. Int. Ed. 2018, 57, 15517

Zhu: Nat. Commun., 2018, 9, 3343.

Three mediated by N-centered radicals, including with boronic acid traps (Ref 35-36) as well as a

Minisci trap (most relevant):

Zhu: Chem. Sci., 2019, 10, 6915

While these previous advances do not preclude publication of this nice new work. It will be important for the authors to put their innovations into the correct context. For example, the O-centered radicals are much more electrophilic, so it is not surprising that their reactivity has been much better developed over several decades, including the recent Minisci trapping reactions. However, the switch to an interrupted HLF mechanism is challenging, and has not been accomplished until recently. Among the 3 reports, 2 use N-F as a radical precursor (Refs 35-36) and the 2019 Zhu work uses hypervalent iodine to access N-centered radicals. In this manuscript, by using an organophotocatalyst to break an N-O bond, the authors have enabled some nice divergent reactivity and selectivity. However, without an explicit introduction to the previous work, this novelty cannot be appreciated fully.

A: We thanks for bringing this relevant works to our attention. Based on these comments and in order to describe the state of the art in remote C(sp³)-H heteroarylation, the introduction is rewritten, especially the second paragraph. Relevant remote C-H heteroarylations by N-radical-triggered 1,5-HAT are described clearly. O-centered radical-mediated processes are also mentioned and cited. The new introduction mainly focuses on the interrupted HLF reactions with heteroarene derivatives, which is quoted as the following:

“Nitrogen-centered radicals (NCRs) are a class of valuable synthetic intermediates, and have become the focus of significant research efforts in recent years.¹¹⁻¹⁴ Radical translocation processes triggered by NCRs, such as 1,5-hydrogen atom transfers (1,5-HAT) demonstrated by the well-established Hofmann-Löffler-Freytag (HLF) reaction, provide an attractive approach for the controllable and selective functionalization of remote inert C(sp³)-H bonds.¹⁵⁻¹⁹ Recently, the photoredox catalysis^{20,21} combined with classic HAT²² provides alternative synthetic tool for remote C(sp³)-H functionalization. This strategy offers a marvelous pathway to selectively achieve mild C-H bond cleavage and C-X (X = halides) and C-N bond formation.¹⁹ Furthermore, interrupted HLF reactions using electron-deficient alkenes,²³⁻²⁷ vinyl boronic acids,^{28,29} allyl sulfones,³⁰ allylic chlorides,³¹ *etc.*^{32,33} as carbon-centered radical traps open a new window for remote C(sp³)-C bond formation. Despite these advances, an interrupted HLF reaction by trapping of the carbon-centered radicals with non-prefunctionalized heteroarenes leading to remote C(sp³)-H (hetero)arylation under photoredox

catalysis is still challenging and remains largely unexplored. Remote C(sp³)–H (hetero)arylation in the assistance of oxygen-centered radical-triggered 1,5-HAT processes has been achieved by several groups.³⁴⁻³⁶ Recently, Zhu et al.³⁷ and Nagib et al.³⁸ independently developed a Cu-catalyzed (hetero)arylation of remote C(sp³)–H bonds with boronic acids as the cross-coupling partners. More recently, Zhu et al.³⁹ reported a hypervalent iodine-promoted remote C(sp³)–H heteroarylation of amides. Inspired by these reports on remote C(sp³)–H (hetero)arylation, we attempted to develop an amidyl radical-triggered, transition-metal free and site-selective remote C(sp³)–H heteroarylation with non-prefunctionalized heteroarenes under photoredox conditions (Figure 1b).”

Q: 4. References: In general, many key references are buried in sentences that do not accurately describe the innovation. Some examples include:

Page 2, line 31 - refers to the Minisci reaction, when it should be a Minisci trap of an intramolecular HAT (Ref 17)

A: This sentence was deleted and relevant references are cited thereafter.

Page 2, line 39 - comparison to conditions for alpha-C-H abstraction in the Minisci reaction are irrelevant, since this work entails HAT from N-centered radicals. Only generation of N-centered radicals would be relevant.

A: All irrelevant seminar works were removed. A comprehensive review is cited instead.

Page 3, top - this should be the key intro about N-centered radicals and the HLF reaction. Also, the difference between Refs 37-48 should be described. Most are interrupted HLF reaction. Ref 44 and 45 are the same. Ref 46 is confusingly the only O-radical HAT, but is not one of the delta arylations mentioned above. If this were the intro paragraph, it would be the correct place to explain the contributions above.

A: These references were reorganized and described as followed: “Furthermore, interrupted HLF reactions using electron-deficient alkenes,²³⁻²⁷ vinyl boronic acids,^{28,29} allyl sulfones,³⁰ allylic chlorides,³¹ *etc.*^{32,33} as carbon-centered radical traps open a new window for remote C(sp³)–C bond formation.” All O-radical HAT chemistry is mentioned and cited thereafter as followed: “Remote

C(sp³)-H (hetero)arylation in the assistance of oxygen-centered radical-triggered 1,5-HAT processes has been achieved by several groups.³⁴⁻³⁶

Q: 5. Regioselectivity: Page 4, first paragraph refers to "intractable regioselectivity issues." This should include a reference. It is also strange because there is no mention of regioselectivity in the optimization, or later on. Does this mean it was >20:1? If so, and given the DFT support, then what are the "intractable regioselectivity issues"?

A: A reference dealing with alkylation of purine derivatives is provided (ref. 43). In this referencing work, by adjusting the amount of tBuOOtBu and reaction time, the selective synthesis of C6-monocycloalkylated or C6, C8-dicycloalkylated purine nucleosides could be realized. In our work, protecting group of N1 position has significant influence on the regioselectivity. The model compound 9-benzyl-9*H*-purine (**2a**) is the optimized substrate and only C6 alkylated purine could be isolated. No C2 or C8 alkylated purine was observed. Initially, 9-methyl-9*H*-purine (**2'a**) was employed instead of **2a**, the C6 alkylated purine (**3'**) was isolated in 76% yield together with 8% yield of C8 alkylated regioisomer. We added this information in Table 1, entry 15. Since C6 seems intrinsically active than C2 and C8, "intractable" is removed.

Q: 6. PCET: Page 6 refers to PCET, but there is no evidence that the proton and electron are coupled in this mechanism. It is just the net loss of an H-atom. It could simply be an asynchronous oxidation and deprotonation. The references cited go to great lengths to determine if those events are coupled, but this paper does not. A simple loss of H-atom would do.

A: We thanks for these insightful comments. As this reviewer indicated, the net result of this process

is the loss of an H-atom ($= \text{H}^+ + \text{e}^-$). Proton transfer (PT) from the carbon atom of the complex **C** to the oxygen atom of the carbonate ion (CO_3^{2-}), and the electron is transferred to the photocatalyst (3CzCIIPN^+). The H^+ and e^- in a single donor (**C**) transfer to two different acceptors ($\text{CO}_3^{2-} + 3\text{CzCIIPN}^+$).

Referring to the literatures, the H-atom ($\text{H}^+ + \text{e}^-$) loss process can proceed via at least three different mechanisms:^[1] (i) HAT, which is a concerted proton-electron transfer from a single donor to a single acceptor; (ii) PCET, which was also a concerned proton-electron transfer, but the proton and electron transfer to (or) from different reagents; (iii) stepwise process involving either initial electron transfer (ET) followed by proton transfer (PT), or PT followed by ET. According to the literature description, this process is not strictly a HAT process, but a PCET or stepwise ET/PT or PT/ET process. Further referred to the application example from Knowles^[51] (the reference in manuscript) and Theopold's^[2] viewpoint "A stepwise ET/PT process is dominant when thermodynamic driving forces for electron transfer from substrate to electron acceptor are only moderately unfavorable. When thermodynamic driving forces for electron transfer are very unfavorable, the concerted proton electron process was significant." In our reaction, the electron transfer process was very thermodynamically favorable (exergonic by 26.1 kcal/mol, see Supplementary Figure 5). In addition, the large amount of base K_2CO_3 (1.0 equiv) was added to this reaction. Studying the oxidation of tryptophan by $\text{Ru}(\text{byp})_3^{3+}$, Meyer^[50] (the reference in manuscript) found that kinetic studies under a variety of conditions have revealed pH regions where ET/PT and PT/ET dominate, when there is a general proton accepting base. Based on the understanding for the above mechanisms and application examples, the PCET mechanism would not be taken into account in this manuscript. We propose this H-atom loss is a stepwise ET/PT or PT/ET process. As this reviewer indicated, the loss of an H-atom is simply to be an asynchronous oxidation and deprotonation, which was supported by our DFT calculations (For details, see our response to Question 8).

The "PCET" in **Figure 2** is corrected to ET and PT.

These works about the above perspective were cited as ref 48 and ref 49, respectively (replacement of 56)

- (1) Warren, J. J., Tronic, T. A. & Mayer, J. M. Thermochemistry of Proton-Coupled Electron Transfer Reagents and its Implications. *Chem. Rev.* **110**, 6961-7001 (2010).
- (2) Gunay, A. & Theopold, K. H. C–H Bond Activations by Metal Oxo Compounds. *Chem. Rev.* **110**, 1060-1081 (2010).

Q: 7. References: The references to seminal work on Marcus Theory and the Fukui Function are a little strange. It would be more helpful to cite recent applications of these tools to C-H functionalization, rather than include definition references that could be found in a textbook. Also, as currently written, the Fukui Index explanation (line 129) is incorrect. The Fukui Index does not “favor” reactivity. Instead, it 'may explain' selectivity. In this manuscript, the authors have correlated their observed reactivity with Fukui values that explain electron density perturbations. This is interesting, but should be stated properly.

A: The references about recent applications of Marcus Theory and Fukui Function are cited as refs 53-55 and refs 57,58 respectively.

Applications of Marcus Theory literatures (replacement of refs 60-64 in manuscript):

- (1) Mayer, J. M. Understanding Hydrogen Atom Transfer: From Bond Strengths to Marcus Theory. *Acc. Chem. Res.* **44**, 36-46 (2011).

- (2) Yang, W., Chen, X. & Fang, W. Nonadiabatic Curve-Crossing Model for the Visible-Light Photoredox Catalytic Generation of Radical Intermediate via a Concerted Mechanism. *ACS Catal.* **8**, 7388-7396 (2018).
- (3) Jones, G. O., Liu, P., Houk, K. N. & Buchwald, S. L. Computational Explorations of Mechanisms and Ligand-Directed Selectivities of Copper-Catalyzed Ullmann-Type Reactions. *J. Am. Chem. Soc.* **132**, 6205-6213 (2010).

Applications of Fukui Function literatures (replacement of ref 66 in manuscript):

- (1) Ma, Y., Liang, J., Zhao, D., Chen, Y.-L., Shen, J. & Xiong, B. Condensed Fukui function predicts innate C–H radical functionalization sites on multi-nitrogen containing fused arenes. *RSC Adv.* **4**, 17262-17264 (2014).
- (2) Cheng, J., Deng, X., Wang, G., Li, Y., Cheng, X. & Li, G. Intermolecular C–H Quaternary Alkylation of Aniline Derivatives Induced by Visible-Light Photoredox Catalysis. *Org. Lett.* **18**, 4538-4541 (2016).

Thank you for your correction and suggestions, the sentence of line 129 was corrected to “The atom with the maximal value of Fukui index is predicted to be the preferred reactive site” in revised manuscript.

Q: 8. The third part of Figure 3 is labeled as PCET. It should just be HAT. Otherwise, there should be DFT support for PCET.

A: According to the description and understanding in our response to Question 6, H-atom loss in this process proceeds via stepwise ET/PT mechanism. In order to verify whether this process is favorable, we computed Gibbs free energy profile for the H-atom loss (Supplementary Figure 5). And the related calculation result, discussion and structure coordinates were added in the revised manuscript and SI.

In the revised manuscript:

“PCET” in Figures 2 and 3 was corrected to stepwise “ET/PT”. And “**Complex II**” and its structure in Figure 3 are replaced by “**E**” and its structure.

In the SI:

From Supplementary Figure 5, the stepwise ET/PT was feasible and kinetically and thermodynamically favorable. The proton transfer process only requires overcoming 1.7 kcal/mol

barrier in PT-ET process. And in the ET-PT process proton transfer was barrierless which can be verified from the decreasing energy with the lengthening C-H distance in Supplementary Figure 6). In contrast with PT, the ET process also was favorable. In the ET-PT process, the electron transfer is exergonic by 26.1 kcal/mol, providing enough driving force for this process. The activation energy in this ET process was estimated to be 6.98 kcal/mol (Supplementary Table 8). Marcus theory is not applicable for describing the rupture and formation of chemical bonds in the electron transfer step. However, in the process from **Complex III** to **3**, besides ET synchronous aromatization occur on the 6-membered pyrimidine ring which can be seen from the average bond lengths in pyrimidine ring of **3** (Supplementary Table 9). Thus, an ET intermediate **F** was designed. And this ET activation energy (9.31 kcal/mol) was estimated to be higher than that (6.98 kcal/mol) in ET-PT process.

Supplementary Figure 5. Computed Gibbs free energy profile for the H-atom loss process in redox neutral coupling reaction of **1a** and **2a** and spin density structure of intermediate **C** and transition state **TS3**. Energies are given in kcal/mol.

Supplementary Table 8. Estimation of the activation barriers for the ET process for the equation (a) and (b) according to Marcus theory.

	$a_1/\text{Å}$	$a_2/\text{Å}$	$R/\text{Å}$	ϵ_{op}	ϵ	$\lambda_0/\text{kcal/mol}$	$\Delta G_r/\text{kcal/mol}$	$\Delta G_{ET}^\ddagger/\text{kcal/mol}$
(a)	8.44	7.90	16.34	2.01	46.83	9.70	-26.15	6.98
(b)	8.44	7.44	15.88	2.01	46.83	10.03	-28.70	9.31

Proton Transfer between E and K_2CO_3 :

Relaxed PES scan along the C-H bond on mixed systems of E and K_2CO_3

Supplementary Figure 6. Relaxed potential energy surface scan along the C-H bond on mixed systems of **E** and K_2CO_3 .

Supplementary Table 9. The selected bond lengths in purine ring of **Complex III** and **3**.

Substance/distance(Å)	C6-N1	N1-C2	C2-N3	N3-C4	C4-N7	N7-C8	C8-N9	N9-C5	C5-C6	C5-C4
Complex III	1.37	1.31	1.38	1.35	1.37	1.37	1.33	1.37	1.44	1.40
3	1.33	1.34	1.33	1.33	1.37	1.37	1.31	1.38	1.41	1.41

Q: 9. Throughout the manuscript and SI, the hydroxamides are referred to as hydroxyamines, but they are amides.

A: Hydroxyamines were corrected to hydroxamides.

Q: 10. Competition: Line 221 refers to “both amide and amine moieties” but both are amidyl radicals. Perhaps reword to the N-alkyl side and the Acyl side?

A: Corrected.

Q: 11. Line 230 states both “kinetically and thermodynamically favored.” Which is it? Also, a 3 kcal/mol difference is shown, but only 2:1 selectivity is observed.

A: Thank you for your careful suggestions and comments. As discussed in the mechanism section of this manuscript, the rate-determining step of this reaction is not the 1,5-HAT process, but the C-C bond formation. So our states in line 230 are incorrect and unconvincing. The computations for the C-C bond formation and stepwise ET/PT steps were added (Supplementary Figure 7 replacement of Figure S1). The 1,5-HAT transition states in Figure 6 are replaced by the C-C bond formation (the rate-determining step) transition states. The C-C bond formation activation barrier is 12.8 kcal/mol in generating product **61**, which is 1.2 kcal/mol lower than the barrier (14.0 kcal/mol) in generating product **62**. The stepwise ET/PT was the preferred process whether in generating **61** or **62**.

Supplementary Figure 7. Computed Gibbs free energy profile for the coupling reaction of **1y** and **2f** generating **61** and **62**, and spin density structure of transition states **TS4**, **TS5**, **TS6** and **TS7**. Energies are given in kcal/mol.

Supplementary Table 10. Estimation of the activation barriers for the ET process of the equations (c)-(f) according to Marcus theory.

	$a1/\text{Å}$	$a2/\text{Å}$	$R/\text{Å}$	ϵ_{op}	ϵ	$\lambda_0/\text{kcal/mol}$	$\Delta G_r/\text{kcal/mol}$	$\Delta G_{ET}^\ddagger/\text{kcal/mol}$
(c)	8.44	8.78	17.22	2.01	46.83	9.19	-29.42	11.14
(d)	8.44	9.20	17.64	2.01	46.83	9.00	-33.84	17.16
(e)	8.44	8.65	17.09	2.01	46.83	9.25	-28.38	9.88
(f)	8.44	7.79	16.23	2.01	46.83	9.77	-34.54	15.70

Reviewer 2:

Q: In this manuscript Yu and coworkers disclosed a Minisci-type alkylation reaction of N-heteroarenes via amidyl radical mediated 1,5-HAT process under organic photoredox catalysis. Different from previous Minisci-type alkylation reactions that requiring acidic conditions, this reaction proceeded well under basic conditions and provided an alternative strategy for the acid sensitive substrates. A broad scope of both hydroxylamines and N-heteroarenes was reported, and functionalization complex bioactive molecules were also demonstrated. The reactivity and regioselectivity were investigated by DFT calculations and the results were consistent with the experimental observation. Overall, this paper well-written and publication in Nature Communication is suggested after addressing some minor issues:

A: We thanks for these insightful comments.

Q: 1. Can the authors speculate on the reason why this Minisci-type alkylation reaction proceeds well under basic conditions?

A: The heteroarenes employed in this work have lower energy of their LUMOs compared to quinoline and pyridine. Therefore, protonation of these heteroarenes is not necessary. Quinoline and pyridine have higher energy of their LUMOs, strong acid is typically used as a stoichiometric additive to protonate the basic heteroarenes and thus lower the energy of their LUMOs to facilitate the radical addition to chemically inert quinoline and pyridine.

Q: 2. What if this reaction is conducted in the acidic conditions using the same starting material and organic photoredox catalysis? Particularly for the unreactive substrates under basic conditions, such as quinoline and pyridine.

A: If this reaction is conducted in the acidic conditions using the same starting material and organic photoredox catalysis, only trace yield of heteroarylation product can be obtained (see Supplementary Table 4, entries 12-13). And when quinoline and pyridine were subjected into our established basic conditions, no target products were detected by GC-MS and TLC.

Q: 3. Recent reports on Minisci-type alkylation via 1,5-HAT process should be cited: Nat. Commun. 2018, 9, 3343; Chem. Sci. 2019, 6915.

A: These two reports have been cited as ref 36 and ref 39, respectively.

Reviewer 3:

Q: The present manuscript describes a novel and facile approach to enabling remote Minisci-type heteroarylations using N-centered amidyl radicals with an interesting mechanism for radical generation under metal-free photo-redox conditions. The novelty of this transformation is that it is a unique example with excellent regioselectivity compared to the well-known Minisci-type alkylation of heterocycles. More than 50 examples and 8 classes of heterocycles (including purines, thiazolopyridines, benzoxazole, benzothiazole, benzothiophenes, benzofurans, thiazoles and quinoxalines) were amenable for this alkylation transformation. Furthermore, a cascade direct C-H bond functionalization of different heteroarenes by taking advantage pH value or polarity of radicals has also been studied in the manuscript. Finally, the investigation of mechanism in the paper is well conducted, and an appropriate radical mechanistic proposal is given based on experimental results, and the site-selectivity and reactivity of this reaction was also well-supported by DFT calculations. Therefore, I suggest this manuscript to be published on Nature Commun.

A: We thanks for these insightful comments.

Q: 1. Authors should include at least one sentence in the introduction section of the manuscript to illustrate the importance of visible light in an organic reaction.

A: We added a statement in the introduction part, saying “Recently, the photoredox catalysis^{20,21} combined with classic HAT²² provides alternative synthetic tool for remote C(sp³)-H functionalization. This strategy offers a marvelous pathway to selectively achieve mild C-H bond cleavage and C-X (X = halides) and C-N bond formation.¹⁹”

Q: 2. The base K₂CO₃ seems quite important for this transformation, the authors should give some comments on it.

A: The acid has negative influence on this reaction. When the reaction proceeds without K₂CO₃ (Table 1, entry 4) or the K₂CO₃ is replaced to the TFA or TsOH, much inferior yields of the target compound was obtained (Supplementary Table 4, entries 12-13). And mechanistically, when the

radical intermediate (**C**) is converted to the product **3**, a deprotonation process is required, which requires the assistance of the base (please refer to Figure 2). Therefore, based on the experimental results and reaction mechanism, the base K_2CO_3 is necessary for this transformation.

Q: 3. When the reaction was irradiation under compact fluorescent light (CFL), is it possible to obtain the product of **3**?

A: When the reaction was irradiation under compact fluorescent light (CFL), only 10% yield of product **3** was isolated. We added this information in Table 1, entry 14 (For more light sources examination, see Supplementary Table 6).

REVIEWERS' COMMENTS:

Reviewer #1 (Remarks to the Author):

All issues have been addressed adequately. This manuscript is now suitable for publication.

Reviewer #2 (Remarks to the Author):

In this revised manuscript, the author has addressed all the issues, and I would like to suggest the acceptance at the current form.

Reviewer #3 (Remarks to the Author):

In their revised version, Yu and co-workers answered the concern of the reviewers and have now an excellent work of high interest in the field of HLF reaction and photoredox catalysis in particular. The authors have carefully rewritten the introduction, added some important references and revised the proposed mechanism, answered all of the questions by the reviewers and adopted the suggestions. The synthetically important paper can now be published immediately as it is.